# Effects of Flap Design on the Periodontal Health of Second Lower Molars after Impacted Third Molar Extraction

**DOI:** 10.3390/healthcare10122410

**Published:** 2022-11-30

**Authors:** Pier Carmine Passarelli, Michele Antonio Lopez, Andrea Netti, Edoardo Rella, Marta De Leonardis, Luigi Svaluto Ferro, Andrea Lopez, Franklin Garcia-Godoy, Antonio D’Addona

**Affiliations:** 1Division of Oral Surgery and Implantology, Department of Head and Neck and Sensory Organs, Fondazione Policlinico Universitario A. Gemelli IRCCS—Università Cattolica del Sacro Cuore, 00168 Rome, Italy; 2Department of Dental Clinic, Dentistry School, Faculty of Biomedical and Health Sciences, Universidad Europea de Madrid, 28670 Madrid, Spain; 3Department of Bioscience Research, College of Dentistry, University of Tennessee Health Science Center, Memphis, TN 38163, USA

**Keywords:** oral surgery, third impacted molar, wound healing, periodontitis

## Abstract

The purpose of this study was to compare the envelope flap and triangular flap for impacted lower third molar (M3) extraction and their effects on the periodontal health of adjacent second molars (M2). A population of 60 patients undergoing M3 extraction with the envelope flap (Group A) or triangular flap (Group B) was analyzed, comparing probing pocket depth (PPD), clinical attachment level (CAL), and gingival recession (REC) recorded at six sites (disto-lingual, mid-lingual, mesio-lingual, disto-vestibular, mid-vestibular, and mesio-vestibular) before (T^0^) and 6 months after extraction (T^1^). There was a statistically significant mean difference in PPD and CAL at two sites, disto-vestibular (dv) and disto-lingual (dl), between values recorded before and 6 months after surgery for either Group A or Group B. Furthermore, for the same periodontal records, at 6 months after surgery, a statistically significant difference was recorded between younger and older patients, implying that the healing process was more beneficial for younger patients. No significant differences were found between the two groups (A and B) in PPDdl, PPDdv, CALdl, and CALdv, confirming that the mucoperiosteal flap design does not influence the periodontal healing process of second molars.

## 1. Introduction

Third molars (M3) are the last permanent teeth to erupt in the oral cavity: they are present in almost 90% of the population [1], and they happen to be the most frequently impacted teeth (with a prevalence of 20–30%, affecting mainly women) [2]. Their removal is one of the most common procedures performed by oral surgeons in their daily practice [3].

The extraction of an impacted lower third molar may determine a wide range of post-surgical consequences: from pain, swelling, mild bleeding, and temporary trismus to transient or permanent paresthesia of the ipsilateral lower lip and possible damage to the adjacent second lower molar [4]. Clinical indications for surgery include frequent pericoronitis, teeth with unrestorable dental caries, cystic development, orthodontic or prosthetic reasons, and periodontal damage to the adjacent second molar (M2) [5,6]. Therefore, the risks and benefits ratio for their removal needs to be carefully evaluated. To minimize potential emergencies, either intra or post-operatory, it is important to conduct an accurate anamnestic investigation to identify conditions that could compromise the surgical procedure, such as a deficit of coagulation factors [7,8].

Several classifications have been developed to assess the risk of complications during mandibular third molar extraction in relation to surgical difficulty. These include those of Winter and Pell and Gregory, which classify the inclinations and positions of third molars based on the relationship among the dental longitudinal axis, the occlusal plane, and the ascending mandibular branch [9].

An impacted M3, proximal to an M2, can lead to asymptomatic periodontal lesions that can, eventually, compromise the mid- and long-term prognosis of these molars [6].

Despite the studies published [10,11,12,13,14], there is no consensus regarding the preventive extraction of impacted M3s as a protective therapy for adjacent M2s.

The extraction of an impacted M3 begins with the elevation of a full-thickness flap to expose the crown of the impacted tooth; several different surgical techniques have been proposed for this application; the main difference between these techniques is whether the flap includes a vertical releasing incision, allowing the surgeon to have a much better exposure of the surgical field and, maybe, obtain more favorable results in terms of the M2 periodontal healing. Two commonly used procedures are the triangular flap and the envelope flap [12,13].

It should be considered that the envelope flap involves direct damage to the intrasulcular vestibular periodontium of M2 that could affect periodontal recovery, whereas in the case of the triangular flap, the periosteal insult of the vertical incision may make the soft tissues less stable during healing, impairing periodontal recovery. The purpose of this study was to evaluate the periodontal status of lower M2s before and after extraction of adjacent impacted M3s, comparing patients treated with envelope or triangular flaps.

## 2. Materials and Methods

Patients who underwent M3 extraction for acute or chronic pericoronitis and periodontal problems at Oral Surgery Unit, Policlinico Agostino Gemelli (Rome, Italy), from January 2017 to November 2017 were retrospectively analyzed to verify compliance with the inclusion criteria. Inclusion criteria were extraction performed using either the envelope or the triangular flap, the distal surface of the mandibular M2 cleaned with an open flap debridement, and availability of a pre-operative and post-operative full-mouth periodontal chart.

Exclusion criteria were represented by missing or root decay of adjacent M2, systemic disease that could affect healing (e.g., diabetes, patients on bisphosphonate therapy, liver disease), uncontrolled periodontal disease, cigarette consumption >10 per day, and no follow-up recall. Patients that did not meet the inclusion criteria were excluded.

In the study, sixty patients, aged from 16 to 81, that had the extraction of a mandibular M3 were included (Table 1). The 60 patients were divided into two groups according to the surgical flap that was adopted: 30 to Group A (envelope flap) and 30 to Group B (triangular flap).

The patients had signed an informed consent to the treatment before dental wisdom tooth extraction, given that the procedure performed is not experimental or innovative, and a consent that the data and findings made could be used for research purposes even in the future. The study was conducted in accordance with the Declaration of Helsinki of 1975, as revised in 2013. Ethical approval was obtained from the Catholic University of Sacred Heart, Rome, with the ethics committee protocol number 23236/17, ID1608. The study was performed following the STROBE Statement guidelines.

The patients were divided into two groups of 30 patients according to the surgical flap that was adopted: Group A (envelope flap) and Group B (triangular flap). As some reports highlighted the possibility that age might affect periodontal healing [15], the study sample was analyzed by distinguishing between two age groups: group 1 was composed of young patients that is under the age of 25 years old (28 patients), while group 2 was composed by patients aged > 25 years (32 patients).

### 2.1. Pre-Surgical Procedure

All patients had the necessary tradiographic examinations (orthopanoramics or CBCT) that were investigated, and the M3 was classified according to the Pell and Gregory and Winter classification [9]. From the clinical charts, the following data related to the adjacent M2 to the extracted M3 were collected at six sites (disto-lingual, mid-lingual, mesio-lingual, disto-vestibular, mid-vestibular, and mesio-vestibular): the probing pocket depth (PPD), the gingival recession (REC), and then the clinical attachment level (CAL). These periodontal indices recorded on the day of extraction (T^0^ = baseline) and 6 months after surgery (T^1^) were retrieved and compared.

Patients had started antibiotic therapy on the day of surgery with amoxicillin + clavulanic acid 1 g (Augmentin 875 mg/125 mg, GSK, London, UK) every 12 h for 5 days, ibuprofen 600 mg (Brufen, Lake Bluff, IL, USA) every 12 h/3 days, and chlorhexidine digluconate mouthwash 0.12% (Curasept ADS, Curaden HealthCare Spa, Saronno, Italy) every 12 h/10 days.

### 2.2. Surgical Procedure

All the M3 extractions were performed by a single trained oral surgeon (P.C.P.). Before the beginning of surgery, all patients rinsed their mouths with 0.2% chlorhexidine (Curasept ADS 0.2%) for one minute. A standard block of the inferior alveolar nerve was performed using 2% mepivacaine with no adrenaline and 2% mepivacaine with 1:100,000 epinephrine (Optocaine, Molteni Dental, Milan, Italy) for the local infiltration of the buccal nerve. All M3 had been extracted after the elevation of a mucoperiosteal flap: for Group A, the envelope flap consisted of a crestal incision extended in the retromolar pad continuing with an intrasulcular one involving the vestibular aspect of the second lower molar, preserving its mesial papilla; for Group B, the triangular flap consisted of a distal crestal incision in the retromolar pad with a mesial relieving vertical incision starting from the transition line between the distal third and the central third of the gingival margin of the M2, connected by an intrasulcular incision (Figure 1). As this was a retrospective study, there was no assignment of patients to groups, but access to the Oral Surgery Unit followed. These are the flaps that we routinely perform, and flap selection was conducted randomly by a coin toss.

Tooth extraction was executed with mild osteotomy and odontotomy with a surgical bur and then with the help of luxating elevators and forceps (Hu-Friedy Mfg. Co.; Chicago, IL, USA). No damage to the M2 nor to the soft gingival tissues occurred. Extraction sockets were cleaned using an alveolar curette (Hu-Friedy Mfg. Co.; Chicago, IL, USA) and physiologic saline, and the distal surface of the mandibular M2 was debrided open flap; then, the socket was packed with a gelatin sponge (Spongostan; Ferrosan Medical Devices A/S, Søborg, Denmark). Primary closure of the surgical wound was achieved using a 3-0 silk non-resorbable suture (Ethilon, Ethicon, Johnson & Johnson Medical Spa, Pomezia, Italy). An analgesic was administered right after extraction (ibuprofen 600 mg: Brufen, Lake Bluff, IL, USA), and ice packs were locally applied for the first day. Sutures were removed after one week after the extraction.

Patients continued the individual maintenance program based on their caries and periodontal risk assessment and were instructed in oral hygiene maneuvers at home. In the follow-up period, patients were revisited at months 1, 3, and 6 after extraction, and no other periodontal treatment was performed on M2.

The PPD of the lower M2s was recorded 6 months after extraction, by another operator (M.A.L.), on 6 points (mesio-vestibular/lingual, disto-vestibular/lingual, centro-buccal/lingual), together with Rec and CAL, using a North Carolina periodontal probe (Hu-Friedy Mfg. Co., Chicago, IL, USA).

### 2.3. Statistical Analysis

The tooth M2 adjacent to the extracted M3 was defined as the statistical unit. The differences between the recorded values at baseline and the same variables recorded after 6 months were defined as the main outcomes. Binomial or discontinuous variables were assessed by means of the chi-square test and Fisher’s exact test. Within-group comparisons were conducted with a paired *t*-test. After analyzing for assumptions, between-group comparisons were conducted with the analysis of covariance (ANCOVA): the independent variable was the “type of treatment”; the dependent variable was the difference between pre-operative measurements and post-operative measurements; the covariables were the age and the preoperative measurements. The analysis of covariance (ANCOVA) was also used to analyze the differences between the two age groups, with “age group” (Groups 1 and 2) being the independent variable, the difference between pre-operative measurements and post-operative measurements being the dependent variable, and the preoperative measurements being the covariable. The threshold of statistical significance was set at 5%. These tests were performed using R Statistical Software (Foundation for Statistical Computing, Vienna, Austria).

## 3. Results

All the molars surgically removed belonged to Pell and Gregory’s class III, distributed as follows: 15 belonging to Pell and Gregory class IIIA, 26 to Class IIIB, and 19 to Class IIIC. According to Winter’s classification, 19 teeth belonged to class 1, 28 to class 2, and 13 to class 3. Preoperative and 6 months after periodontal values of each group are presented in Figure 2.

In Group A, there was a statistically significant mean difference of 1.7 mm and 1.9 mm, in PPD at two sites (disto-vestibular and disto-lingual) between values recorded before and after 6 months since the procedure (*p* < 0.05) and a statistically significant difference of 1.43 mm and 1.67 mm in CAL at two sites (disto-vestibular and disto-lingual) between values recorded before and after 6 months (*p* < 0.05) (Table 2).

In Group B, there was a statistically significant mean difference of 2.7 mm and 2.67 mm in PPD at two sites (disto-vestibular and disto-lingual) between values recorded before and after 6 months since the procedure (*p* < 0.05) and a statistically significant mean difference of 2.3 mm and 2.43 mm in CAL at two sites (disto-vestibular and disto-lingual) between values recorded before and after 6 months (*p* < 0.05). There was no significant difference for the other outcomes (Table 2).

These differences (Preoperative value − Postoperative value), defined as Delta (∆) PPDdv (disto-vestibular periodontal probing depth), ∆PPDdl (disto-lingual periodontal probing depth), ∆CALdv (disto-vestibular clinical attachment level), and ∆CALdl (disto-lingual clinical attachment level) are reported and compared between the two treatment groups in Table 3. There was no statistically significant difference between the two groups for any of the recorded variables (ANCOVA, *p* > 0.05).

These differences were also compared between the two age groups: significantly higher ∆PPDdv (Young = 2.519 ± 1.76, Old = 1.93 ± 1.27), ∆PPDdl (Young = 2.593 ± 1.76, Old = 2.03 ± 1.26), ∆CALdv (Young = 2.333 ± 1.57, Old = 1.485 ± 1.39), and ∆CALdl (Young = 2.481 ± 1.78, Old = 1.697 ± 1.45) was recorded in younger patients than in older patients (Table 4). A statistically significant difference was found between the two age groups for each recorded variable, indicating that an M3 extraction in younger patients (age ≤ 25) has a better outcome on the adjacent M2 periodontal status than in older patients (age > 25) (ANCOVA, *p* < 0.001).

Figure 3 shows the results of the analysis of covariance (ANCOVA) described in the Methods.

## 4. Discussions

Several studies have been carried out to investigate the effect of lower M3s extraction on the periodontal health of the adjacent M2, using different surgical techniques [16,17,18,19,20,21]. In the present study, a significant improvement of PPD and CAL both on the disto-vestibular and on the disto-lingual aspect of M2s in both groups was recorded. The present findings, therefore, conflict with other reports reporting that extraction of M3 can damage the periodontal attachment of M2, leaving a residual periodontal defect [11,12,22] and instead support the theory that the extraction of an impacted M3 can be beneficial for the periodontal health of the M2, which has already been described by several authors [23,24,25,26,27,28].

The presence of impacted M3s, even if they are asymptomatic, might represent an important risk factor for the periodontal health of M2s. This possibility should be carefully evaluated during the clinical decision-making process regarding retention or extraction, especially when M3s are non-functional or when their removal will not affect the patients’ occlusal function [16,17]. As stated by Richardson et al. [14], the removal of an impacted M3 should be carefully evaluated in patients with an adjacent M2 with good periodontal health as the surgical procedure could worsen its conditions, whereas surgery might improve them in patients presenting increased PPD levels, regardless of the type of incision performed leading to a better periodontal healing [28].

Data from the present study indicate that an M3 extraction in younger patients (age ≤ 25) has a significative better outcome on M2 adjacent periodontal status than in older patients (age > 25). According to the present findings and to several others [8,29], these extractions should preferably be performed as soon as the clinical condition of the M3 warrants it. Performing this procedure in younger patients is indeed accompanied by more favorable healing and, therefore, a better long-term periodontal prognosis for the second molar. The current findings support the ones by Rosa et al. [21], stating that there is no difference in the periodontal healing process of M2s among different types of mucoperiosteal flaps. In agreement with Chaves et al. [27], the type of mucoperiosteal flap should be chosen according to the surgeon’s preferences rather than on the assumption that the periodontal health of M2s would improve on this basis. As reported in previous studies, flap design in lower M3 surgery influences primary wound healing but does not seem to have a lasting effect on the health of the periodontium on the distal aspect of preceding second molars [19]: different surgical techniques do not show different outcomes in terms of PPD reduction and CAL gain [18]. Based on the present results, the flap design should be chosen according to the surgeon’s preferences, also evaluating adequate exposure and visibility of the surgical field in complex extractions, such as total bone inclusions and abnormal tooth root anatomy [30]. An impacted M3 can potentially cause a large periodontal defect on the adjacent M2 as its distal surface cannot be properly cleaned by the patient, and it is a constant source of inflammation [12,23,25,31]. In this study, an improvement in PPD and CAL was found in both flap designs, probably related to the accurate open debridement of the distal surface of M2 during the third molar extraction procedure, in agreement with data also reported by other studies [23,25].

Limitations of the current study include the retrospective setting, which does not allow us to rule out possible procedural bias; a further limitation is the length of follow-up: it would be interesting to evaluate the long-term periodontal data as part of a prospective study.

Another limitation concerns the variability of M3’s position, the influence of which cannot be excluded. Strengths of the study include the fact that the treatments were performed by a single expert clinician and that the data were evaluated and analyzed separately by other authors.

## 5. Conclusions

There was a statistically significant mean difference in PPD and CAL at two sites (disto-vestibular and disto-lingual) between values recorded before and after 6 months after third molar extraction in both the envelope flap and triangular flap. On the other hand, there was no statistically significant difference in third molar extractions between the use of an envelope flap and a triangular flap regarding PPD, CAL, and REC of the second molar. In addition, data from this study indicate that third molar extraction in younger patients (age ≤ 25) has a significantly better outcome on the periodontal status of the adjacent second molar than in older patients (age > 25). Based on these data, a triangular or envelope flap design may not be preferred for surgical extraction of the lower third molars to improve periodontal outcomes of the adjacent second molar.

## Figures and Tables

**Figure 1 healthcare-10-02410-f001:**
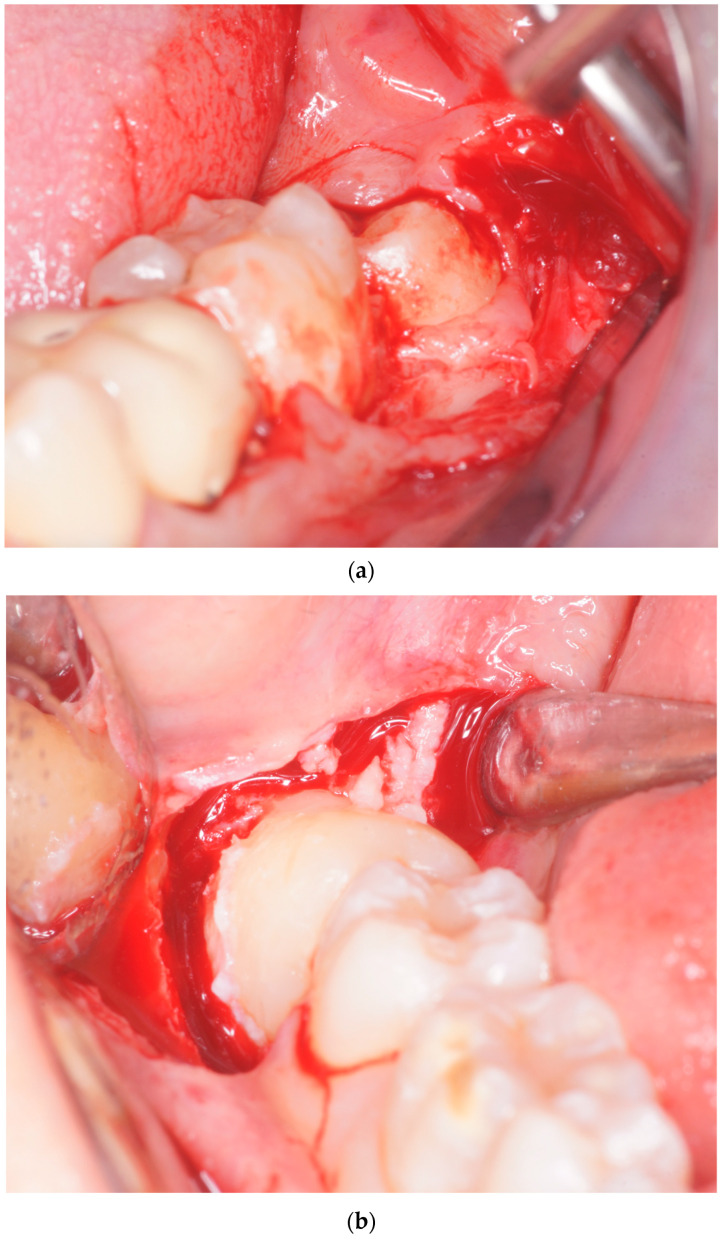
(**a**) Surgical area exposed with envelope flap: crestal incision extended in the retromolar pad continuing intrasulcular to the vestibular aspect of M2. (**b**) Surgical area exposed with triangular flap: distal crestal incision in the retromolar pad with a mesial relieving vertical incision starting from the transition line between the distal third and the central third of the gingival margin of the M2.

**Figure 2 healthcare-10-02410-f002:**
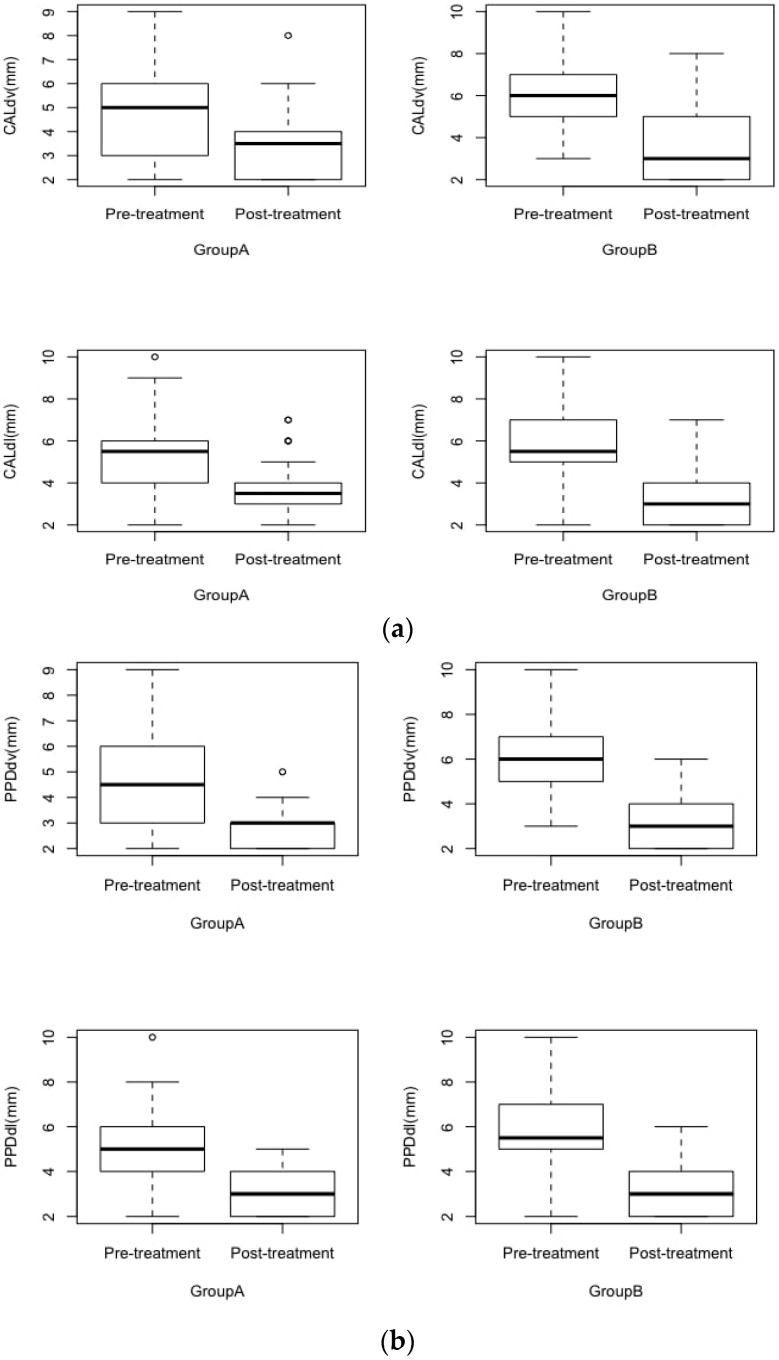
Periodontal values at baseline and 6 months after third molar extraction in Group A (envelope flap) and Group B (triangular flap). (**a**) CAL in groups A and B, before and after extraction at the distovestibular and distolingual points. Box plot: median, upper quartile, lower quartile, upper extreme, lower extreme. CALdv = CAL measurement at the disto-vestibular aspect of teeth; CALdl = CAL measurement at the disto-lingual aspect of teeth. (**b**) PPD in groups A and B, before and after extraction at the distovestibular and distolingual points. Box plots: median, upper quartile, lower quartile, upper extreme, lower extreme. PPDdv = PPD measurement at the disto-vestibular aspect of teeth; PPDdl = PPD measurement at the disto-lingual aspect of teeth.

**Figure 3 healthcare-10-02410-f003:**
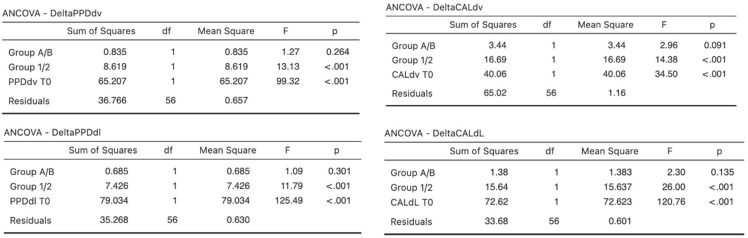
Between-group comparisons with ANCOVA: treatment type (A/B) as the independent variable; difference between preoperative and postoperative measurements as the dependent variable; covariables were age (1/2) and preoperative measurements. Legend: CALdv = CAL measurement at the disto-vestibular aspect of teeth; CALdl = CAL measurement at the disto-lingual aspect of teeth. PPDdv = PPD measurement at the disto-vestibular aspect of teeth; PPDdl = PPD measurement at the disto-lingual aspect of teeth; T^0^: preoperative measurements.

**Table 1 healthcare-10-02410-t001:** Demographical analysis of the sample.

	Group A	Group B	*p*-Value
Mean Age	37.73 ± 17.45	35.7 ± 19.98	0.6 *
Sex	16 M, 14 F	13 M, 17 F	0.6 **
Smoking	25 N, 5 Y	24 N, 6 Y	1 ***

* Welch *t*-test, ** Chisquare test, *** Fisher’s exact test.

**Table 2 healthcare-10-02410-t002:** Comparison of periodontal values (mean + standard deviation) preoperatively and 6 months after of each group.

	Group A				Group B			
	T^0^		T^1^		T^0^		T^1^	
	dv	dl	dv	dl	dv	dl	dv	dl
**PPD** (mm)	4.63 ± 1.61	5.07 ± 1.78	2.93 ± 0.8 **	3.17 ± 0.95 **	5.9 ± 1.71	5.9 ± 1.96	3.2 ± 1.21 **	3.27 ± 1.28 **
**CAL** (mm)	5.03 ± 1.77	5.4 ± 1.92	3.6 ± 1.52 **	3.73 ± 1.51 **	5.97 ± 1.73	5.93 ± 2.18	3.67 ± 1.77 **	3.5 ± 1.63 **
**REC** (mm)	0.37 ± 0.67		0.67 ± 0.92 **		0.23 ± 0.50		0.47 ± 0.82 *	

Paired *t*-test within group: ** *p*-value < 0.001 * *p*-value < 0.05; T^0^: preoperative values; T^1^ after 6 months values; dv: disotvestibular; dl: distolingual.

**Table 3 healthcare-10-02410-t003:** Comparison of ∆ values (mean + standard deviation) between treatment groups: Group A (envelope flap) and Group B (triangular flap).

	dv		dl	
	Group A	Group B	Group A	Group B
∆PPD (mm)	1.7 ± 1.37	2.7 ± 1.53 ^#^	1.9 ± 1.49	2.67 ± 1.47 ^#^
∆CAL (mm)	1.43 ± 1.5	2.3 ± 1.44 ^#^	1.67 ± 1.63	2.43 ± 1.59 ^#^
∆REC (mm)	−0.3 ± 0.53	−0.23 ± 0.5 ^#^		

ANCOVA between groups: ^#^ *p*-value > 0.05; ∆PPD = difference between PPD values at T^0^ and at T^1^; ∆CAL = difference between CAL recordings at T^0^ and at T^1^; ∆REC = difference between REC recorded at T^0^ and ad T^1^; dv = distovestibular; dl = distolingual.

**Table 4 healthcare-10-02410-t004:** Comparison of ∆values (mean + standard deviation) between young (Group 1) and old patients (Group 2).

	dv		dl	
	Group 1	Group 2	Group 1	Group 2
∆PPD (mm)	2.519 ± 1.76	1.93 ± 1.27 **	2.593 ± 1.76	2.03 ± 1.26 **
∆CAL (mm)	2.333 ± 1.57	1.485 ± 1.39 **	2.481 ± 1.78	1.697 ± 1.45 **

ANCOVA between groups; ** *p*-value < 0.001; ∆PPD = difference between PPD values at T^0^ and at T^1^; ∆CAL = difference between CAL recordings at T^0^ and at T^1^; dl = distolingual.

## Data Availability

The data presented in this study are available on request from the corresponding author.

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
