# Peer review of "Effects of Flap Design on the Periodontal Health of Second Lower Molars after Impacted Third Molar Extraction"

_healthcare, 2022, doi:10.3390/healthcare10122410_

Round 1

Author Response

Dear Reviewer,

Thank you for your work in revising our manuscript. Your comments have allowed us to better clarify the suggested aspects and improving the quality of the manuscript. We have corrected and reorganized the manuscript. As the corresponding author, I have provided a point-by-point response (green text) and highlighted in green the parts added in the text.

Best Regards

Dear Authors,
These are my suggestions/queries
1. Title of manuscript is rather lengthy

  • We have modified it as suggested

Introduction
1. Dear authors, kindly comment as the need/ purpose of the study can be emphasized a bit further. As the flap design forms a factor for evaluation, kindly consider elaborating
further

  • As suggested we further elaborated on the assumptions underlying the purposes

Methodology
2. It is mentioned that- Patients underwent M3 mandibular molar extraction at our clinic ..
kindly revise as who underwent instead. Kindly mention the no: of patients considered too

  • We have added the causes of extraction and the number of patients considered, as also suggested in later comments
  1. Kindly comment on below sentences
    a. All patients gave their informed consent for inclusion at the study before they participated.
    b. from January 2017 to November 2017 were retrospectively analyzed to verify compliance with the inclusion criteria
    kindly comment whether the participants gave their consent for the treatment procedure that was conducted prior to the study as it is stated that the study was retrospectively analyzed which includes evaluation of the flap design and the periodontal features of M2. Kindly verify and revise accordingly
  • Dear reviewer, thank you for requesting this clarification on an unclearly expressed issue. The patients signed an informed consent to the treatment (dental wisdom tooth extraction), given that the procedure performed is not experimental or innovative, and a consent that the data and findings made could be used for research purposes even in the future. We have also modified it in the text
  1. It is mentioned that- All patients had performed radiographic examinations by
    orthopanoramics or CBCT.. kindly revise
  • We modified and specified that in any case the patients had the necessary preoperative radiographic investigations.
  1. Dear authors, kindly comment on the design of the study. If retrospective in nature,where the study details were analyzed from the available records, kindly revise the narration of the methodology accordingly or was it a prospective interventional clinical study design. Kindly verify, comment and revise and restructure the methodology narration

- Dear Reviewer, thank you for your observation. We modified the verbs, and all surgical and clinical treatments were described using the past perfect, which frames a tense prior to that of the analysis

  1.  It is mentioned that- All teeth were extracted after elevation of a mucoperiosteal flap: for Group A the.. kindly verify and revise as M3
  • We modified it as correctly suggested.
  1. Methodology needs to be narrated in past tense. kindly verify and revise
  • As previously reported, we modified the verb forms to place the surgeries before the analysis
  1. It is mentioned that- Patients agreed to continue the individual treatment program based on their caries and periodontal risk assessment and were instructed in oral hygiene maneuvers at home.
    Dear authors, kindly comment whether the above program mentioned continued and after that the patient were reviewed for follow up data. kindly comment whether any of the treatments were subjected to M2 too. If yes, whether it will influence the outcome or the results that were to be recorded. kindly comment whether it might lead to bias
  • Dear Reviewer, thank you for the clarifying request. We corrected the “treatment” in “manteinance”. Patients were stabilized based on risk assessment, and no further periodontal treatment was performed on M2. We have also added it in the text.
  1. It is mentioned that- the study sample was divided into two age groups: group A was composed by young patients that is under the age of 25 years old, while group B was composed by old patients (Age > 25).. kindly consider mentioning the former in the
    methodology instead
  • Thank you for your observation. The age groups are group 1 (< 25 years) and group 2 (> 25 years). We have included them in the methodology.

Results
1. It is mentioned that- In the study sixty patients, aged from 16 to 81 that had the extraction of a mandibular M3 were included (Table 1). The 60 patients were divided into two groups according to the surgical flap that was adopted: 30 to Group A (envelope flap) and 30 to Group B (triangular flap).
Dear authors, kindly comment. If the study is retrospective in nature, the above details can be included in the methodology as specific criteria in selection process would have been drawn out. Kindly comment

  • Dear reviewer, as suggested, we have moved these descriptive data into the methodology.

Discussion
1. It is mentioned that- The presence of M3s, even if they are asymptomatic, might
represent an important risk factor for the periodontal health of M2s.. kindly comment
whether it is impacted or erupted M3 or both

  • The mention is of the M3 impacted. We have specified as suggested.
  1. Kindly refer as present/ current study instead of our in the narration
  • We have modified as suggested.

Reviewer 2 Report

This article is well-organized and edited. The article discussed the effect of the triangular flap and envelope flap designed on the second molar's periodontal health after extracting the adjacent impacted tooth. The results showed some significant information to oral surgeons. From an oral surgeon's point of view,  there are still some factors related to the post-extracted M2 periodontal status, not only because of the surgical flap design. The position of the impacted tooth and alignment of the tooth also showed significant effects on the periodontal status of the M2 tooth. How could the authors exclude the influence of the M3 position? Did the author find the influence of surgical size and incision length with the different flap designs? 

Author Response

 This article is well-organized and edited. The article discussed the effect of the triangular flap and envelope flap designed on the second molar's periodontal health after extracting the adjacent impacted tooth. The results showed some significant information to oral surgeons. From an oral surgeon's point of view,  there are still some factors related to the post-extracted M2 periodontal status, not only because of the surgical flap design.

- Dear Reviewer,

Thank you for your work in revising our manuscript. Your comments have allowed us to better clarify the suggested aspects. I have provided a point-by-point response and highlighted in gray the parts added in the text.

The position of the impacted tooth and alignment of the tooth also showed significant effects on the periodontal status of the M2 tooth. How could the authors exclude the influence of the M3 position?

- Dear reviewer,

thank you for the certainly important clinical observations. The only way would have been to select cases with the same inclusion classification and location, but the sample would have been very small given the low probability. We will include this evaluation among the limitations of the study.

Did the author find the influence of surgical size and incision length with the different flap designs?

- Thank you for your very insightful observation. For the same incision length, the triangular flap allows more surgical area to be exposed, so it is preferred for deeper inclusions, as reported in lines 378-380. To have more extension with the envelope flap it would be necessary to extend the intrasulcular incision.

Reviewer 3 Report

This is a retrospective study comparing two different treatments and the healing outcomes. The authors claimed they followed the STROBE statement guidelines. The materials and methods are described in detail.

I have some comments for the authors.

1. Figure 1a and 1b show four figures, the author needs to clarify the difference between the upper one and the lower one.

2. The authors conducted ANCOVA to analyze the difference of post-treatment. Authors should use standard format (ANCOVA table) to show the difference between the two groups. The result description of table 3 did not show "Table 3" in the paragraph. 

3. The authors did not show the participants number in groups. Only describe there are 60 patients in total.

4. The statistical method should be labeled on the table footnote.

Author Response

 This is a retrospective study comparing two different treatments and the healing outcomes. The authors claimed they followed the STROBE statement guidelines. The materials and methods are described in detail.

            Dear Reviewer,

Thank you for your work in revising our manuscript. Your comments have allowed us to better clarify the suggested aspects, improving the quality of the manuscript. We corrected and reorganized the manuscript and tables. As the corresponding author, I have provided a point-by-point response and highlighted in light blue the parts added in the text.

Best Regards

I have some comments for the authors.

  1. Figure 1a and 1b show four figures, the author needs to clarify the difference between the upper one and the lower one.

- We have specified it in footnotes, as suggested.

  1. The authors conducted ANCOVA to analyze the difference of post-treatment. Authors should use standard format (ANCOVA table) to show the difference between the two groups. The result description of table 3 did not show "Table 3" in the paragraph. 

- We have reorganized the tables so that they contain the essential information and are more easily consulted. As suggested, we have added the missing reference to the table in the text (it is now table 4).

  1. The authors did not show the participants number in groups. Only describe there are 60 patients in total.

           - We have specified it as suggested.

  1. The statistical method should be labeled on the table footnote.

          -  We have added it as suggested.

Reviewer 4 Report

The manuscript is well written and the topic is interesting but some things can be improved; the Authors must:

1) Include in Materials and Methods some images of the two flaps to better understand the differences;

2) Specify why a patient is included in the group A or in the group B (on the basis on what did you decide to perform the envelop or the triangular flap in a patient?)

3) Provide images of the outcomes of local region in the two groups to understand the differences.

Author Response

The manuscript is well written and the topic is interesting but some things can be improved;

 Dear Reviewer,

Thank you for your work in revising our manuscript. Your comments have allowed us to better clarify the suggested aspects, improving the quality of the manuscript. We have added required information in the manuscript. As the corresponding author, I have provided a point-by-point response and highlighted in light yellow the parts added in the text.

Best Regards

the Authors must:

1) Include in Materials and Methods some images of the two flaps to better understand the differences.

           - We have added them as suggested

2) Specify why a patient is included in the group A or in the group B (on the basis on what did you decide to perform the envelop or the triangular flap in a patient?)

- These are the flaps that we routinely perform, and unless specifically   indicated, it is a random choice. We have added it in the text.

3) Provide images of the outcomes of local region in the two groups to understand the differences.

- We do not have clinical images of healings, only survey data; being a retrospective evaluation, they were not planned. Instead, we possibly have images of radiographic follow-ups to evaluate CAL. Those of the surgical stages, however, are performed more routinely.

Round 2

Reviewer 1 Report

The revisions are satisfactory. Please do a thorough spell check 

Author Response

Dear Reviewer,

Dear Reviewer,

We will evaluate the spell check with the journal's editing service. Thank you again for your time and suggestions that improved the manuscript.

Best Regards

Reviewer 3 Report

Please find a professional biometrician to help with the table format.

Author Response

Dear Reviewer,

as suggested we have added the ANCOVA table standard format. We are very grateful for your recommendations to improve the manuscript.

Best Regards